# Dipotassium Glycyrrhizininate Improves Skin Wound Healing by Modulating Inflammatory Process

**DOI:** 10.3390/ijms24043839

**Published:** 2023-02-14

**Authors:** Camila dos Santos Leite, Gabriel Alves Bonafé, Oscar César Pires, Tanila Wood dos Santos, Geovanna Pacciulli Pereira, José Aires Pereira, Thalita Rocha, Carlos Augusto Real Martinez, Manoela Marques Ortega, Marcelo Lima Ribeiro

**Affiliations:** 1Laboratory of Immunopharmacology and Molecular Biology, São Francisco University Medical School (USF), Bragança Paulista, São Paulo 12916-900, Brazil; 2Laboratory of Cell and Molecular Tumor Biology and Bioactive Compounds, São Francisco University Medical School (USF), Bragança Paulista, São Paulo 12916-900, Brazil; 3Laboratory of Pharmacology, Taubaté University (UNITAU), Taubaté, São Paulo 12030-180, Brazil; 4Department of Surgery and Proctology, São Francisco University (USF), Bragança Paulista, São Paulo 12916-900, Brazil; 5Postgraduate Program in Biomaterials and Regenerative Medicine, Faculty of Medical Sciences and Health, Pontifical Catholic University of São Paulo, São Paulo 05014-901, Brazil

**Keywords:** skin wound healing, inflammation, DPG, animal model, rats

## Abstract

Wound healing is characterized by a systemic and complex process of cellular and molecular activities. Dipotassium Glycyrrhizinate (DPG), a side product derived from glycyrrhizic acid, has several biological effects, such as being antiallergic, antioxidant, antibacterial, antiviral, gastroprotective, antitumoral, and anti-inflammatory. This study aimed to evaluate the anti-inflammatory effect of topical DPG on the healing of cutaneous wounds by secondary intention in an in vivo experimental model. Twenty-four male Wistar rats were used in the experiment, and were randomly divided into six groups of four. Circular excisions were performed and topically treated for 14 days after wound induction. Macroscopic and histopathological analyses were performed. Gene expression was evaluated by real-time qPCR. Our results showed that treatment with DPG caused a decrease in the inflammatory exudate as well as an absence of active hyperemia. Increases in granulation tissue, tissue reepithelization, and total collagen were also observed. Furthermore, DPG treatment reduced the expression of pro-inflammatory cytokines (*Tnf-α*, *Cox-2*, *Il-8*, *Irak-2*, *Nf-kB*, and *Il-1*) while increasing the expression of *Il-10*, demonstrating anti-inflammatory effects across all three treatment periods. Based on our results, we conclude that DPG attenuates the inflammatory process by promoting skin wound healing through the modulation of distinct mechanisms and signaling pathways, including anti-inflammatory ones. This involves modulation of the expression of pro- and anti-inflammatory cytokine expression; promotion of new granulation tissue; angiogenesis; and tissue re-epithelialization, all of which contribute to tissue remodeling.

## 1. Introduction

Wound healing is characterized by a systemic and complex process of cellular and molecular activities that occur naturally, aiming to repair the affected region anatomically and functionally. This event is composed of four basic phases: hemostatic, inflammatory, proliferative, and remodeling or maturation, which involve cell division, proliferation, migration, neovascularization, synthesis of protein elements, contraction, re-epithelialization, and structural remodeling of the injured tissue [1,2,3]. After tissue injury, multiple responses (vasoconstriction, platelet buffering, and coagulation) are triggered to restore local hemostasis [2,4]. Within 24 h, an inflammatory state characterized by vasodilation, increased vascular permeability, tissue edema, neutrophil migration, macrophage activation, and the release of pro-inflammatory cytokines sets in at the site [5,6,7]. In this period, cytokines, interleukins, and chemokines [8] stimulate the synthesis of other pro-inflammatory mediators [9,10], as well as the release of prostaglandins, increasing the inflammatory condition [11]. Subsequently, starting on the third or fourth day, a new tissue (granulation) begins to form simultaneously with neovascularization and tissue re-epithelialization [1,2,12], an event that can last for up to 2 or 3 weeks. Finally, after the second or third week, the wound tends to regress, undergoing tissue remodeling and final repair. However, this event can last for months or years, depending on the initial damage and the persistence of stimuli [1,2,13]. 

Natural compounds, especially those derived from various plant species, have been used successfully in studies on the treatment of cutaneous wounds and inflammatory changes in animal models [14,15]. Licorice is the common name for the perennial herbaceous plant *Glycyrhiza glabra*, which is a member of the Fabaceae family and is native to Asia and southern Europe. It has been studied since ancient times for a better understanding of the nutritional and pharmacological properties present mainly in its roots. These roots are rich in several biological compounds, such as saponins [16,17], estrogens, phytosterols, coumarins, vitamins [18,19], chalcones, and flavonoids [16,17,20]. Glycyrrhizin (G), the main triterpenoid saponin derived from licorice, found in about 41.84–114.33 mg/g [21], is known to give rise to glycyrrhizic acid (GA), a compound that has antitumor, antiallergic, antiviral, antibiotic, and anti-inflammatory properties [16,22,23,24,25]. In addition, dipotassium glycyrrhizinate (DPG), a side product derived from GA, has antiallergic, antioxidant, antibacterial, antiviral, anti-inflammatory [26,27,28], and antitumor [29,30] effects. More recently, it has been described as having gastroprotective [22] and healing [31] properties related to DPG.

It has been described that the anti-inflammatory action of DPG refers to its inhibitory effect on the enzyme hyaluronidase, which is involved in damage to the extracellular matrix and is responsible for the release of histamine from granules present in mast cells, as well as that of inflammatory mediators such as leukotrienes and prostaglandins (PGs) [26,27,28,32]. This effect is similar to those of corticosteroids, but does not induce the side effects (i.e., erythema, hyperhidrosis, or allergic skin reactions) observed with the use of these drugs [26,27,28].

Although a recent study found that rats treated with DPG experienced effective epidermal proliferation and re-epithelialization, as well as an increase in type I collagen levels [31], the mechanism of action of this chemical on other crucial stages of wound healing has hardly been investigated. Thus, this work aimed to evaluate the effects of DPG on wound healing in rats after 3, 7, and 14 days of treatment, addressing the involvement of pro- and anti-inflammatory factors.

## 2. Results and Discussion

### 2.1. DPG Attenuates Inflammation, Promoting Formation of New Granulation Tissue and Tissue Re-Epithelialization

Wound healing simultaneously involves distinct cellular types in its hemostatic, inflammatory, proliferative, and tissue remodeling phases [2]. Thus, it is important to emphasize that the speed and efficiency of wound repair, as well as scar formation, are closely related to a balance between inflammatory and proliferative responses [33,34].

According to the macroscopic observations, it was noted that both the treated groups (DPG) and the untreated groups (Control) showed tissue re-epithelialization at the end of the 14th day of experimentation. Although the wounds in the DPG-treated groups were apparently smaller, there were no significant differences between the groups when compared at days 1, 3, 7, and 14 (day 1, DPG vs. Control (*p*-value = 0.2630); day 3, DPG vs. Control (*p*-value = 0.1924); day 7, DPG vs. Control (*p*-value = 0.4871) and; day 14, DPG vs. Control (*p* = 0.8460)) of experimentation, respectively (Figure 1A,B). These results support a prior study in which the authors noted progressive healing of the excisional incision following 7, 14, and 21 days of DPG treatment [31].

Physiologically, the inflammatory process manifests singular characteristics, such as inflammatory exudate (vasodilation, increased permeability, and recruitment of leukocytes) and active hyperemia (congested vessels and reddish tissue coloration) [6]. The data obtained in this study indicate that there was a trend towards reduction of the inflammatory exudate after treatment with DPG, suggesting a potential anti-inflammatory effect of DPG (Figure 2A,E). At the third and seventh days of treatment, there was a trend toward decreased active hyperemia in the DPG-treated groups when compared to the control group. However, on day 14, hyperemia was significantly absent in the DPG group (*p*-value 0.05). Although no significant differences were initially observed between the groups, the total absence of active hyperemia in the animals treated on day 14 may suggest that the DPG promoted an anti-inflammatory effect (Figure 2B,E). These findings agree with other studies and suggest that the DPG may have promoted a more accentuated reduction in vasodilation, vascular permeability, leukocyte infiltration, release of pro-inflammatory mediators, and, consequently, inflammation, thereby inducing an improvement in subsequent cell proliferation [6,7]. Additionally, glycyrrhizin, a compound similar to DPG, was demonstrated to be able to inhibit NO synthesis and decrease inflammation, increase cell proliferation, and speed up wound healing in vitro by a subsequent study evaluating anti-inflammatory, antioxidant, and proliferative activities [35].

On the third day, we noticed the absence of granulation tissue in both groups, which is physiologically normal during this period. On the seventh day, it was significantly higher in the DPG group (*p*-value < 0.05). On the 14th day, the granulation tissue was similar in both groups (Figure 2C,E). Once the inflammatory process is over, the path towards reconstruction of the damaged tissue continues, and intense cell proliferation and migration take place. During the proliferative phase, new tissue (granulation) begins to form simultaneously with angiogenesis and tissue re-epithelialization [1,2,12]. In general, the formation of granulation tissue composed of fibroblasts, COL-III, new vessels, scattered macrophages, and VEGF-permeated connective tissue begins between the fourth and fifth day after the initiation of tissue healing [36,37]. In our study, we observed that there was a greater amount of granulation tissue in the treated animals around the seventh day, a phase of intense cell proliferation. The use of collagen-containing microcapsules containing glycyrrhizin has also been shown to increase hydroxyproline content, capillary, and fibroblast proliferation in granulation tissue, as well as to increase and uniform collagen fiber formation, promoting neovascularization and wound healing in rats [38].

On day 3, there was an absence of re-epithelialization in both groups, which is physiologically normal during this period. This tissue begins to form between the sixth and fourteenth day after the start of tissue repair. On the seventh day, this characteristic showed a tendency to increase in the DPG group. However, on day 14, the increase in re-epithelialization was significantly greater in the DPG group (*p*-value 0.0001) (Figure 2D,E). From the seventh day of healing, depending on the characteristics of the wound, re-epithelialization tends to occur. It is characterized by the migration of keratinocytes and epithelial cells from the wound margins toward the center and the subsequent formation of skin appendages [37,39]. Keratinocytes stimulated by keratinocyte growth factor (KGF) [40,41] repair the epithelial layer [36,42]. Taken together, our findings suggest that DPG may positively influence cell proliferation and migration in the dermis and epidermis, and, thus, the scar. Consistent with previous studies, more effective epidermal proliferation and re-epithelialization were observed during the proliferative phase of skin healing in rats treated with DPG for 7 days [31]. Additionally, rats given glycyrrhizin showed decreased inflammation, accelerated re-epithelialization at day 8, better tissue remodeling, and enhanced collagen deposition at day 16 [43].

Concerning total collagen, at days 3 and 7, although there was no significant difference, a relatively higher density of total collagen content was observed in the DPG group. On day 14, it was observed that treatment with DPG promoted a significant increase in total collagen levels (*p*-value < 0.05) (Figure 3A,B). Similar improvements in healing were seen during skin wound remodeling in DPG-treated rats, as was an increase in the presence/distribution of type I and III collagens [31]. Furthermore, histological examinations of the wound sites in rats treated with licorice revealed an increase in collagen synthesis/deposition as well as re-epithelialization. [44].

During proliferation, the deposition and organization of a provisional matrix rich in collagen, mainly type 3, also occur [45]. Fibroblasts initially synthesize greater amounts of COL-3 and, later, COL-1 [45], with 80–90% of the collagen present in the normal reticular dermis generally being type 1, arranged in intertwined and organized fibers [46]. Subsequently, during wound contraction, the collagen is arranged perpendicular to the wound edges, while the area to be re-epithelialized is reduced. In the course of proliferation and remodeling of the tissue, the myofibroblasts contract, reapproximating the edges of the wound, and the provisional ECM is replaced by a more resistant tissue, which determines features of the skin’s integrity and the final quality of the scar [45,47]. In agreement, we saw greater type I collagen expression during the 14-day tissue remodeling phase, primarily in the DPG-treated group, indicating a more robust recovery.

### 2.2. DPG Modulates the Expression of Pro- and Anti-Inflammatory, Proliferative, and Remodeling Genes

In the present work, we also evaluated the effects of DPG (C_42_H_60_K_2_O_16_) in pro-inflammatory (*Cox-2, Tnf-α, Nf-kb, Il-1α, Il-6, Il-8, Irak-2* e *Hmgb-1*), anti-inflammatory (*Il-10*), angiogenic (*Vegf*), and structural remodeling (*Col-1*) genes, mainly to understand its effect on the wound repair of rat skin.

In response to tissue injury, several pro-inflammatory factors are synthesized and secreted by neutrophils, macrophages, and other cells, as well as anti-inflammatory mediators and growth and structural factors, which act in the different phases of wound healing [45]. At first, platelets, leucocytes, and endothelial cells release inflammatory mediators, mainly COX-2 [48,49], IL-1 e IL-8 [11,50,51], TNF-α, IFN-γ, CXCL1, CXCL8 [52,53], and adhesion molecules [6,7]. These mediators and others recruit large numbers of inflammatory cells [54,55,56], mainly neutrophils [52,53], to the wound site, which aim to degrade the damaged matrix and remove the damaging agent, preventing infection [53,57].

Following the above, in the present study, the gene expression levels of several pro-inflammatory markers were investigated. We discovered that DPG treatment had an immediate effect, significantly lowering *Tnf-α* expression (Figure 4A). Similarly, we also observed that on the third and seventh days, there was a significant decrease in the expression levels of *Cox-2*, *Il-8,* and *Irak2* in the DPG groups (Figure 4B–D). In the case of *Nf-kb* and *Il-1*, we observed mRNA repression after 7 days, as well as after 7 and 14 days of DPG treatment (Figure 4E,F, respectively). Finally, DPG significantly increased the expression of *Il-10* after the third, seventh, and fourteenth days (Figure 4G). Il-10 has been demonstrated to diminish neutrophil and macrophage infiltration at the site of injury, as well as the release of pro-inflammatory cytokines (Il-1b, Il-6, and TNF-α), when it is present in infiltrating epidermal and mononuclear cells [58]. A murine full-thickness wound model has shown that the use of IL-10 (ovIL-10) is effective in reducing macrophage infiltration and suppressing pro-inflammatory mediators, which accelerate granulation tissue formation and re-epithelialization and enhance wound revascularization, positively regulating skin repair [59]. As a result, we saw that DPG administration raises Il-10 levels, lowers the expression of pro-inflammatory cytokines, and reduces inflammatory infiltration. Our data show that DPG helps to improve re-epithelialization by lessening the inflammatory process overall.

Specifically, TNF-α, which is constitutive in cutaneous tissue [8,60], is released mainly by neutrophils, lymphocytes, mast cells, macrophages, and keratinocytes during the inflammatory response [6]. It is also able to induce and control the inflammatory process through binding to TNFR1-p55 and TNFR2-p75, triggering pro-inflammatory signaling cascades [61]. TNF-α activates and regulates in-loop NF-kB, a factor that regulates the transcription of genes encoding pro-inflammatory cytokines and also mediates keratinocyte survival and proliferation [62]. Its signaling rapidly and transiently activates the transcription of NF-kB-dependent target genes, such as IL-1β, IL-6, and IL-8 [63,64]. In fact, we observed accentuated *Tnf-α* and *Nf-kb* reductions on the third and seventh days, respectively (Figure 4A–E). DPG also promoted significant increases in *Il-10* expression (Figure 4G) and in the total collagen content during the three evaluated periods (Figure 3A), suggesting an apparent anti-inflammatory and proliferative mediator’s modulation. Our findings also suggest that DPG decreased *Tnf-α* expression on the third day (Figure 4A), which may contribute to the significant reductions in *Nf-kb* and *Il-8* on the seventh day of treatment (Figure 4C–E). TNF-α, IL-1, IL-6, and NF-kB levels were significantly reduced in rats treated for 3, 7, and 14 days with lupeol, a natural triterpene found in olive, fig, mango, carrot, soybean, melon seed, and grapes. In contrast, increased IL-10 levels were verified. As a result, the NF-kb pathway was disrupted, resulting in decreased inflammation, angiogenesis, and improved collagen proliferation and deposition in all three exposure periods [65]. It has been demonstrated that DPG was able to drastically lower the expression levels in a distinct biological setting. *Tnf-α, Il-1β,* and *Il-6,* through *Hmgb1* inhibition in vitro, reduced intestinal epithelial inflammation and the severity of colitis in animals [66]. Additionally, DPG can reduce the renal inflammatory process by blocking *Tnf-α* and *Il-1β*, in vitro and in vivo, through the NF-kB and MAPK inhibitory pathway [67].

Furthermore, it is known that moderate levels of TNF-α favor skin recovery by recruiting immune/inflammatory cells, contributing to the normal transition between the inflammation and cell proliferation phases [33,68]. On the other hand, a drastic reduction in leukocyte infiltration and expression of pro-inflammatory cytokines, in addition to excessive angiogenesis, less re-epithelialization, and fibrous tissue formation, was reported in mice with TNFR1-p55 inactivation [68]. In contrast, excess TNF-α retards the proliferation of keratinocytes and fibroblasts, leading to persistent inflammation and delayed skin healing [33,68]. We observed a repression, but not a suppression, of *Tnf-α* during the inflammatory phase, which remained stable during proliferation in the treated groups (Figure 4A). It is suggested that DPG might possibly balance the expression of *Tnf-α* and dependent cytokines, promoting anti-inflammatory effects without impairing the physiological transition between the inflammatory and proliferative phases. In a psoriasis-like murine inflammatory model and in epidermal keratinocytes (HaCaT), it has been demonstrated that glycyrrhizin can reduce inflammation and enhance healing through regulation of Tnf-α-induced ICAM-1 expression via NF-kB/MAPK signaling [69].

NF-kB also has a role in the regulation of homeostasis in response to inflammatory stimuli, expressing cytokines [62], cell adhesion molecules, and growth factors during healing [70]. NF-kB acts primarily through activation of the canonical IkB/NF-kB signaling pathway [71,72]. Similar to TNF-regulation, the persistence of NF-kB activation can induce chronic inflammation on the one hand, but on the other, its suppression also causes negative changes and, in some cases, inflammation [73]. Thus, it is clear that a careful balance between TNF-α and NF-kB activation and inhibition is required for the maintenance of cellular homeostasis as well as the induction and resolution of skin inflammation during healing, as observed in our study. The repression, but not suppression, of *Nf-kb* seen in treated animals (Figure 4E) suggests that DPG may possibly promote an anti-inflammatory effect by balancing the expression of Nf-kb and dependent cytokines. Accordingly, it has been demonstrated in a study utilizing glycyrrhizic acid, another licorice derivative, that this substance can quicken wound healing in a mouse skin model. Mechanistically, it has been discovered that the NF-kB signaling pathway mediates increased cell migratory activity and inhibition of the inflammatory process [74].

COX-2 can modulate PGE2 synthesis during the inflammatory phase by mediating TNF-α/IL-8 signaling pathways [6,75], such as MAPKp38/NF-kB/AP-1 [76,77,78], among others, thus increasing the inflammatory status. It was reported that in rat wounds, COX-2 was expressed mainly in the basal layer of the epidermis, peripheral hair follicle cells, and fibroblast-like cells and capillaries around the wound 12 h after injury, reaching a peak at day 3 [79]. Subsequently, a gradual decrease in COX-2 was reported between days 3, 5, 7, and 14, as well as high levels on day 3 and a gradual decrease on days 7 and 14 in curcumin-treated rats and controls, respectively [80]. Moreover, in a murine model of a pressure ulcer, it was reported that the selective COX-2 inhibitor celecoxib can repress iNOS, Cox-2, and PGE2 and, consequently, the inflammatory process, promoting cell differentiation and re-epithelialization with improved healing [81]. Similarly, in our study, we observed significantly decreased expression of *Cox-2* mRNA on days 3 and 7, which remained lower on day 14 in the animals treated with DPG. In control animals, expression gradually increased between days 3 and 7 and decreased on day 14 (Figure 4B). These findings suggest that the repression of *Cox-2* by DPG in the initial phase of the repair allows the tissue to recover more rapidly, since it diminishes the synthesis of PGE, thus reducing inflammation.

The inhibition of Cox expression using Cox-1 and 2 siRNA or ibuprofen, as well as by diclofenac, caused a reduction in PGE2 and VEGF release in HaCaT cells and mouse excisional wounds, respectively, leading to a consequent impairment of neovascularization, both in vivo and in vitro [82]. On the other hand, we found a likelihood of a balance in *Vegf* expression in the treated groups (Figure 4H), suggesting that DPG, in addition to remarkably reducing the inflammatory process, may have also promoted positive and early effects on vascular proliferation (neoangiogenesis). In another study, COX-2 inhibition by topical hesperidin hydrogels accelerated dermal regeneration in mice, resulting in early wound contraction and reducing mean healing time by 5–7 days, with increased collagen [83]. Conversely, COX-2 overexpression can stimulate aberrant inflammatory and fibrogenic responses, leading to severe inflammation [84], and can exacerbate fibroblast proliferation and collagen synthesis, leading to abnormal healing [85], events that could be suppressed by DPG.

Other cytokines, such as IL-8, also act mainly in the acute phase of the inflammatory process [6,86,87]. IL-8 is expressed by leukocytes, monocytes, and macrophages, and to a lesser extent by fibroblasts, endothelial cells, and keratinocytes [87]. It is secreted in response to inflammatory stimulation, primarily through the induction of IL-1, TNF-, and IFN-, and it functions by binding to CXCR1, CXCR2, IL-8R, and IL-8RA on inflammatory and endothelial cells [51,88]. High levels of IL-8 and TNF-α are known to increase the secretion of other pro-inflammatory cytokines, exacerbating inflammation [89,90,91] and decreasing the proliferative and migratory capacities of epithelial cells and fibroblasts, thus directly contributing to delayed healing and the induction of chronic wounds [92]. In our study, we found that DPG significantly reduced *Il-8* expression during the inflammatory and proliferative phases (Figure 4E), as well as Tnf-α expression during the inflammatory phase (Figure 4A). It is suggested that DPG suppresses the expression of *Il-8* and, consequently, of other cytokines induced by these, mainly in inflammatory cells.

In immune and inflammatory cells, IRAK-2-mediated IL-1/TRL signaling controls inflammation [93,94]. IRAK-2 activity is critical for the TRL and/or IL-1R signaling pathways [95], promoting early activation of NF-kB and induction of inflammatory mediators [96]. In skin inflammation, IRAK-2 acts mainly on keratinocytes, triggering the regulation of an alternative epidermal differentiation pathway through effects on the epidermal differentiation-associated transcription factor (TF-ZNF750) [90], which enhances the immune response, thus elevating the pro-inflammatory condition [97,98]. In the present study, we observed a suppression in the expression of Irak-2, mainly during the inflammatory and proliferative phases, in treated animals (Figure 4D), suggesting a possible modulation of the DPG on some of the signaling pathways mentioned above. IRAK-2-deficient mice have been shown to synthesize reduced levels of pro-inflammatory cytokines [99], and they are more resistant to inflammatory processes and septic shock [100,101]. There are few studies on IRAK-2 in skin healing, and little is known about the function of its signaling in the epidermis under healthy and/or inflammatory conditions. However, the relationship of IRAK-2 activation to the severity of chronic inflammatory skin diseases (psoriasis and dermatitis cutanea) [97] and tumor progression [93] is clear. Therefore, as previously mentioned, the *Irak-2* suppression observed in the treated animals suggests that relevant activity of DPG is involved in the attenuation of cutaneous inflammation through negative modulation of *Irak-2* (Figure 4D).

Furthermore, during the inflammatory phase, inactive IL-1 is induced by TLR/TNF activation or IL-1 receptor activation by active IL-1 or IL-1 [101,102]. IL-1 can stimulate the secretion of other acute phase cytokines, COX-2, and PGEs through activation of the ERK1/2, MAPKp38 and JNK signaling pathways [103,104,105]. In addition, it can stimulate inflammation via the MAPK/AP-1 pathways [106] and IL-1/NF-kB [105] in fibroblasts and the Myd88/TRAF6/NF-kB pathway in epidermal stem cells [107]. In our study, there was similar low expression of *Il-1* in treated and control animals at day 3, but at days 7 and 14, this expression was remarkably repressed by DPG.

At the end of the inflammatory phase and during the transition to the proliferative phase, the inflammatory mediators are important in promoting the synthesis and activity of the inflammatory cells that previously predominated, gradually providing more room for the anti-inflammatory factors that regulate or re-establish physiological balance and the subsequent intense proliferation [6,7,108]. In the meantime, macrophages (M1) give way to macrophages (M2) [109], which degrade the remaining neutrophils, eliminating inflammation, and act in the transition to proliferation [110,111]. Monocytes, macrophages (M2), lymphocytes, and epidermal cells release growth factors [6,112,113] and anti-inflammatory cytokines [114]. These cytokines, mainly IL-10, attenuate skin inflammation by directing subsequent angiogenesis, granulation, re-epithelialization, and regeneration [114,115], in addition to helping to prevent fibroproliferative disorders [114,116]. Skin IL-10 expression has been shown to reduce TNF- and IL-1 mRNA expression in vitro [112] and in vivo [117], as well as to attenuate the inflammatory response via the IL-10R/STAT3 signaling pathway, regulating TLR4/NF-kB in dermal fibroblasts and reducing fibroblast differentiation, ECM deposition, collagen network contraction, and hypertrophic scar formation [112,117]. In our study, we verified an accentuated and significant increase in the expression of *Il-10* in all periods (3, 7, and 14 days) (Figure 4G), mainly during the inflammatory phase, as well as a reduction in *Tnf-α* and *Nf-kb* in treated animals (Figure 4A–E). It is suggested that a positive modulation of the DPG leads to the induction of anti-inflammatory genes, which, consequently, may help to achieve a balance between inflammatory response and resolution and tissue proliferation and reorganization, leading to the formation of a more uniform scar without signs of cutaneous fibrosis.

We also evaluated the effect of treatment on the expression levels of the marker *Vegf*. The data obtained in this work indicate that initially, at 3 days, DPG induces the expression of this gene. At 7 days, however, repression by treatment was observed. On day 14, no significant differences were observed between the two groups (Figure 4H). VEGF is one of the most important pro-angiogenic factors. It is secreted by endothelial cells, macrophages, keratinocytes, and fibroblasts in this period [42,118], and removes pro-angiogenic effects through binding to FLT-1/VGFR1, FLK-1/KDR/VEGFR2, and FLT-4/VEGFR3 receptors [118,119]. Studies on the expression pattern of VEGF in skin healing in mice have reported that mRNA *Vegf* and VEGF levels increase in the first 24 h [120,121], rising gradually on days 3 and 5 and returning to normalization between the 7th [121,122] and 14th days [121]. In our study, we observed a significant increase in *Vegf* expression on day 3, with a tendency to gradually decrease between days 7 and 14 in treated animals. These findings suggest that initially, the DPG may have stimulated early angiogenesis by inducing a greater release of VEGF during the inflammatory phase (Figure 4H). Subsequently, it may have contributed to maintaining the balance of Vegf synthesis by fibroblasts and keratinocytes during cell proliferation, resulting in better tissue recovery and scar formation without evidence of fibrosis. In rat wounds, collagen and glycyrrhizin microcapsules have been demonstrated to up-regulate Vegf and miRNA-21, encouraging neovascularization [38]. In rat wounds treated with licorice extract, improved healing was also seen, along with increased angiogenesis and collagen deposition, which was mediated by the upregulation of *Vegf*, *Fgf*, and *Tgf* expression levels [44].

Ultimately, we evaluated the effects of treatment on the expression of *Col-1*. The data presented in this work indicate that after only 14 days of treatment with DPG, an increase in the expression of this gene was observed (Figure 4I). As already mentioned, during the wound remodeling phase, the COL-3-rich transitional ECM is replaced by a fibrillar, interwoven, and organized COL-1-based scar [45,122,123,124]. It has been reported that increased Col-1 expression and hydroxyproline levels were found during proliferation and remodeling (7 and 14 days) and inflammation and remodeling (3 and 14 days), respectively, in the healing of diabetic rats treated with calcium alginate [125]. Additionally, it was shown that the application of amitriptyline-based nanoparticles (Amitrip) to diabetic rat wounds could quicken tissue remodeling by increasing hydroxyproline levels and collagen deposition due to up-regulation of Vegf and Col-1 [126]. In our study, we observed a higher level of synthesis of total collagen in animals treated during proliferation and remodeling (7 and 14 days) (Figure 3A), as well as a higher level of expression of *Col-1* in animals treated during remodeling (14 days) (Figure 4I). These data suggest that the DPG, in addition to inducing an increase in total collagen, may also have more precisely balanced the transition from COL-3 produced initially to COL-1, which is predominant in the mature scar, without inducing fibrosis. In agreement, it has been demonstrated that using injectable hydrogels loaded with DPG (HP-3/DPG10) promotes efficient tissue remodeling in mice by decreasing inflammation, promoting quick wound healing, and increasing collagen deposition. Additionally, although the literature suggests a link between elevated collagen synthesis and the development of tissue fibrosis, this fact is more closely linked to an escalation of the inflammatory process [127,128,129,130] and collagen production, particularly type III collagen [130,131]. We observed rapid inflammatory process resolution, improved levels of total collagen, and significant expression of type I collagen during wound remodeling, indicating the maturity and resilience of the new tissue. However, collagen III levels have been found to increase in the deep dermis of hypertrophic scars and keloids [130,131].

## 3. Materials and Methods

### 3.1. DPG Gel Cream

Dipotassium glycyrrhizinate (C_42_H_60_K_2_O_16_) was manipulated as a cream at a concentration of 2%. It was supplied by Verdi Cosmetics (register number: 64.786.031/0001-00, Joanópolis, Brazil) (Table 1).

A cream gel was prepared using water, Sepigel 305, Dipotassium Glycyrrhizinate, and Euxyl PE 9010. Procedure: Dipotassium Glycyrrhizinate (2 g) was dissolved in water (93.6 mL). Next, Sepigel 305 (4 g) was added, aiming at controlling the viscosity of the formula. Finally, the preservative Euxyl PE 9010 (0.4 mL) was added. All ingredients were constantly mixed until complete homogenization. The prepared cream gel was stored in a clean, dry container and kept in a common refrigerator until use throughout the experimental period. The cream gel was used for topical application onto the wounds for 14 consecutive days during the experiment. We emphasize that the compound was used only for experimentation in this study, and is not currently marketed for the same purpose [31,132].

### 3.2. Animals and Experimental Groups

The research was previously approved by the Ethical Committee for Animal Use (CEUA) of São Francisco University (protocol n° 007.06.2020), following the guidelines of the Brazilian Society of Laboratory Animal Sciences (SBCAL). Wistar rats were supplied by Laboratory Animals Breeding and Commerce (ANILAB; Paulínia, Brazil) and kept at the Animal Experimentation Vivarium of São Francisco University in individual cages, at 22 ± 3 °C on a 12 h light/dark cycle, with free access to a standard diet and ad libitum water.

A total of 24 male rats (Wistar) were used, for a sample of 24 adult animals (3 months old, weighing between 274 and 300 g). The animals were randomly divided into 6 groups (n = 4, each), where they constituted untreated groups (Control) and treated groups (DPG): animals that were not treated (control groups, 3, 7, and 14 days) and animals that were treated with DPG (DPG groups, 3, 7, and 14 days). We want to be clear that all of the animals had surgery on the same day (day 0).

### 3.3. Excision Wound Model

The animals were subjected to an anesthetic procedure with 2% xylazine hydrochloride (Xilazin^®^, Syntec, Santana de Parnaíba, Brazil) (10 mg·Kg^−1^) associated with 5% dextrocetamine hydrochloride (Ketamin^®^, Cristália, Itapira, Brazil) (25 mg·Kg^−1^), prepared by combining 0.5 mL of xylazine (10 mg) with 0.5 mL of ketamine (25 mg) to a volume of 1.0 mL, which was administered intraperitoneally (1.0 mL·Kg^−1^).

After being anesthetized, the respective animals were positioned on appropriate tables in the horizontal prone position, and skin antisepsis with chlorhexidine 2% (RIOHEX^®^, Rioqumica, So José do Rio Preto, Brazil), followed by 0.5% alcoholic chlorhexidine (RIOHEX^®^, Rioqumica, So José do Rio Preto, Brazil), was performed as part of the pre-surgical preparation [132]. With the help of a scalpel (handle and blade number 15), Mezenbaum scissors curve, and anatomical forceps, each animal—having already been identified by its group—had one circular excision of the skin made in the median plane of the dorsal region, which was constrained in depth by the muscular aponeurosis. This was accomplished by precisely measuring the 2 cm diameter of each excision using a caliper and a plastic circular mold (Universal Digimess 100003) [132]. Thereafter, animals were housed individually and monitored in properly disinfected cages to prevent infection or further damage to the wounds after recovering from anesthesia.

### 3.4. Topical Treatment

Wounds in the treated groups (DPG) were treated once a day for 3, 7, and 14 days at the same time and by the same researcher. This was performed with topical application of 2% DPG cream (in the amount of 0.1 mL/each animal) using cotton-tipped flexible shafts (Cotonetes^®^—Johnson & Jonhson, São Paulo, Brazil) and without the use of bandages. For this purpose, 0.1 mL of cream was measured using an insulin-type syringe (Slip without needle—Injex^®^, Ourinhos, Brazil). Treatment started 24 h after the surgical procedure in the treated groups (DPG). The wounds in the untreated groups (Control) did not receive any intervention during the entire experimental period (3, 7, and 14 days) (Figure 5).

### 3.5. Collection, Storage, and Processing of Samples

Then, according to each experimental group, skin samples (scar tissue) were excised and gathered in due time (3, 7, and 14 days) for histological and gene expression analyses. For this, the animals, under the effect of the 3% isoflurane anesthetic, were submitted to active euthanasia by sodium thiopental (Thiopentax^®^, Cristália, Itapira, Brazil) at a dosage of 100 mg·Kg^−1^ [133].

For microscopic analysis, the skin samples were fixed in 10% formaldehyde solution (Labsynth^®^, Diadema, Brazil) for 24 h, fixed at the extremities in cork, dehydrated in increasing concentrations of ethanol (Labsynth^®^), clarified in xylene (Labsynth^®^), embedded in paraffin (Labsynth^®^), and submitted to microtomy (Lupetec MRPO3, São Carlos, Brazil). The slides (5 μm thick sections) were deparaffinized in two xylene baths (10 min each) hydrated in decreasing concentrations of ethanol (100%, 95%, 80%, and 70%) and distilled water, and stained with hematoxylin–eosin (HE) for semi-quantitative analysis of inflammatory parameters (inflammatory exudate and active hyperemia) and proliferative parameters (granulation tissue and reepithelialization). The modified and adapted inflammatory/proliferative score scales were used, as well as Masson’s Trichrome (MT) for quantitative analysis of the total collagen. After staining, the slides were dehydrated in increasing concentrations of ethanol (70%, 80%, 95%, and 100%) and xylene, then mounted with synthetic balsam from Canada.

The samples collected for gene expression analysis were identified and stored in individual tubes for processing and analysis: in RNAlater stabilization solution at room temperature for 24 h, in a refrigerator for another 24 h, frozen at −20 °C for 1–2 weeks, and, finally, frozen at −80 °C. Additionally, the following practices were used for the collection: The entirety of each animal’s scar tissue was removed, and from each sample taken, a tiny flap was taken for gene expression analysis without including any potential crusts.

### 3.6. Macroscopic Analysis

Wound areas were measured on days 1, 3, 7, and 14 of experimentation with the aid of a pachymeter (Universal Digimess 100003) [132]. Wound areas were measured (large versus small diameter). Subsequently, from the values found, the diameter of the circumference of each wound was calculated using the formula (C = 3.14 × r^2^).

### 3.7. Microscopic Analysis—Inflammatory and Proliferative Scores

Through the modified and adapted inflammatory score grading scale, the following scores were assessed for microscopic anatomopathological analysis (semi-quantitative): histological characteristics related to the inflammatory process (presence of leukocyte infiltration/inflammatory exudate and active hyperemia) and characteristics related to tissue proliferation (presence of granulation tissue and re-epithelialization).

Hematoxylin–eosin (HE) was used for semi-quantitative analysis of inflammatory parameters (inflammatory exudate and active hyperemia) and proliferative parameters (granulation tissue and re-epithelialization) through the modified and adapted inflammatory/proliferative score scales and with Masson’s Trichrome (MT) for quantitative analysis of total collagen [134]. Briefly, for the HE technique, the slides were stained with hematoxylin for 5 min, washed in running water, stained with eosin for 3 min, and, finally, washed in running water. For the MT technique, the slides were kept for 1 h in Bouin’s solution in the oven, cooled, washed in running water until completely clear, washed in distilled water 3 times, and treated with Biebrich’s Scarlet solution for 3 min. Again, they were washed in running distilled water, kept in phosphotungstic acid for 8 min, washed in running distilled water, kept in aniline blue for 3 min, and, finally, washed in running water once again.

Each of the parameters (inflammatory exudate, active hyperemia, granulation, and re-epithelialization) were stratified as: 0 = absent; 1 = mild; 2 = moderate; and 3 = intense, according to the changes found using HE (Table 2 and Table 3) [134]. Each parameter in each experimental group was analyzed separately by an experienced collaborating pathologist who was unaware of the experimental groups to which the animals belonged. For this purpose, the slides (HE) were analyzed in 3 different fields, and the value adopted for each parameter analyzed for each animal in the group was the mean value found after reading three different fields of the lesion area. The final value assigned to each sample analyzed in each experimental group was the mean value obtained by adding the values of each parameter [134]. The 500× final magnification data were used for the analysis.

The total collagen content was analyzed by quantitative reading on the slides in three different fields. The computer-assisted image analysis program was used. The selected image was captured by a video camera previously coupled to an optical microscope (Eclipse DS50—Nikon Inc., Osaka, Japan) and then analyzed by the program NIS-Elements (Nikon Inc., Japan) [134]. By means of color histograms, the software determines the color intensity of each area selected for measurement, transforming the chosen color into a numerical expression for each selected field of view. Using the color histogram in the RGB (red, green, blue) system, the blue color was selected, the intensity of which was captured by the number of pixels containing the color and then converted into a numerical value. The final value considered for each measured field of each sample was represented by the average of the values found after the evaluation of three different fields [134]. For both analyses, a final magnification of 500× was used.

### 3.8. RNA Extraction and Reverse Transcription Quantitative PCR (qPCR)

A portion of the lesion samples from each animal was collected in RNAlater and frozen at −80 °C. Total RNA was extracted from skin tissues using Trizol^®^ reagent (Applied Biosystems, Foster City, CA, USA) following the manufacturer’s protocol. After extraction, ~100 ng of RNA was used for cDNA synthesis using the High-Capacity cDNA Archive Kit (Applied Biosystems), following the manufacturer’s recommendations.

q PCR was performed using a 7300 real-time PCR System (Applied Biosystems), and threshold cycle numbers were determined using RQ Study Software (Applied Biosystems). Reactions were performed in triplicate, and threshold cycle numbers were averaged. The reaction mixture was prepared using Power Up SYBR^®^ Green Master Mix (Applied Biosystems). The reaction was cycled with preliminary Uracil–DNA glycosylase was treated for 2 min at 50 °C with a denaturation step for 2 min at 95 °C, followed by 45 cycles of denaturation at 95 °C for 15 s, annealing for 15 s, and primer extension at 72 °C for 15 s. This was followed by melting point analysis of the double-stranded amplicons consisting of 40 cycles of 1 °C decrement (15 s each) beginning at 95 °C. The first derivative of this plot, dF/dT, is the rate of change of fluorescence in the reaction, and a significant change in fluorescence accompanies the melting curve of the double-stranded PCR products. A plot of –dF/dT vs. temperature displays these changes as distinct peaks. *Cox-2, Tnf-α, Nf-kb, Il-1α, Irak-2, Il-8rb, Il-10, Vegf, Col-α1,* and *18s* (Table 4) expressions were examined and normalized to a constitutive gene (*18s*).

### 3.9. Statistical Analysis

Data are expressed as the mean ± S.E.M. Comparisons among groups of data were made using one-way ANOVA followed by the Dunnett Multiple Comparisons test. An associated probability (*p*-value) of <5% was considered significant.

## 4. Conclusions

Our findings have led us to the additional conclusion that DPG reduces inflammation by promoting skin wound healing through modulation of various mechanisms and signaling pathways, such as: anti-inflammatory, through modulation of pro- and anti-inflammatory cytokine expression; promotion of new granulation tissue, angiogenesis, and tissue re-epithelialization, indicated by modulation of VEGF and stimulation of collagen synthesis; and contributing to tissue re-epithelization. Our research supports topical DPG’s ability to reduce inflammation and promote healing, suggesting that this substance has potential as a therapeutic agent, particularly for cutaneous inflammatory disorders and illnesses.

## Figures and Tables

**Figure 1 ijms-24-03839-f001:**
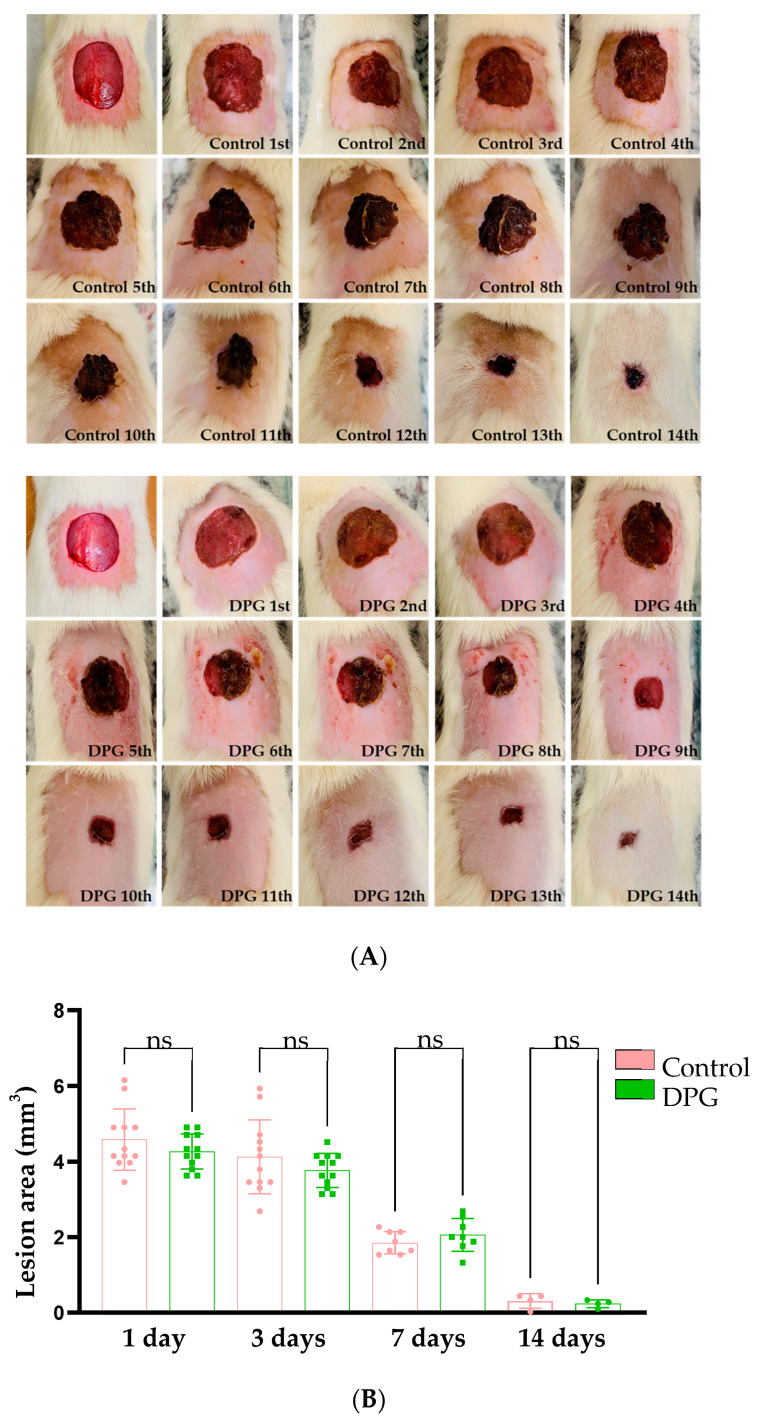
(**A**) Representative macroscopic images of wound healing areas of untreated adult animals (Control—left panel) and those treated with DPG (DPG—right panel) throughout the experimental period (0 to 14 days); (**B**) Quantitative analysis of the measurements of wound healing areas on days 1, 3, 7, and 14, ns—non significant.

**Figure 2 ijms-24-03839-f002:**
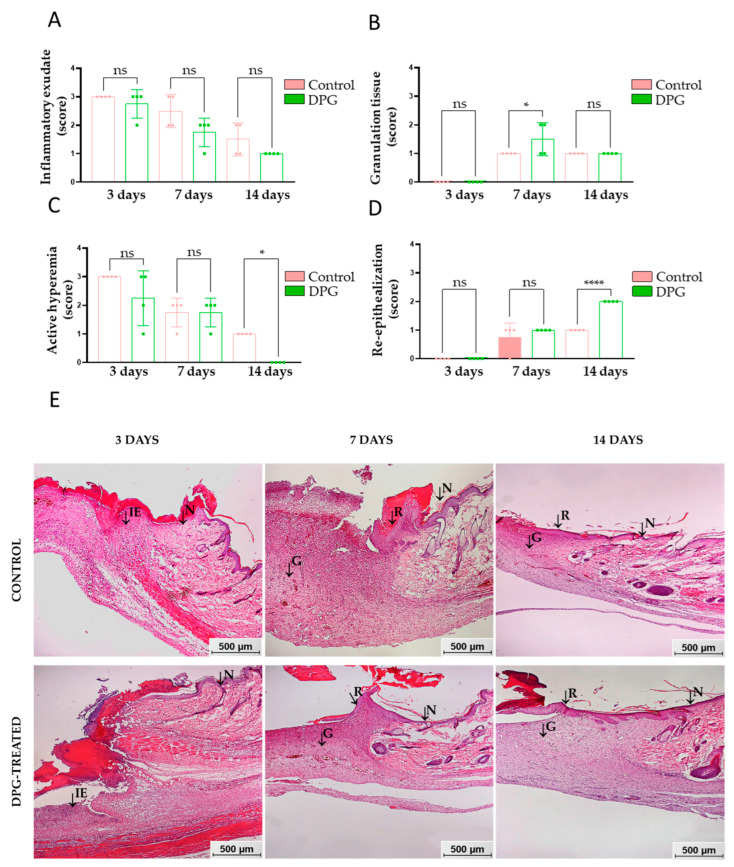
Semi-quantitative characteristics of wound healing areas at days 3, 7, and 14 of experimentation. (**A**) Inflammatory exudate; (**B**) active hyperemia, DPG vs. Control, day 14 (* *p* < 0.05); (**C**) granulation, DPG vs. Control, day 7 (* *p* < 0.05); (**D**) re-epithelialization, DPG vs. Control, day 14 (**** *p* < 0.0001); (**E**) representative histological images of the wound healing areas of the animals. Note the presence of normal tissue (N), inflammatory exudate (IE), granulation tissue (G), and re-epithelialization tissue (R). HE staining: 50× magnification (bar = 500 μm).

**Figure 3 ijms-24-03839-f003:**
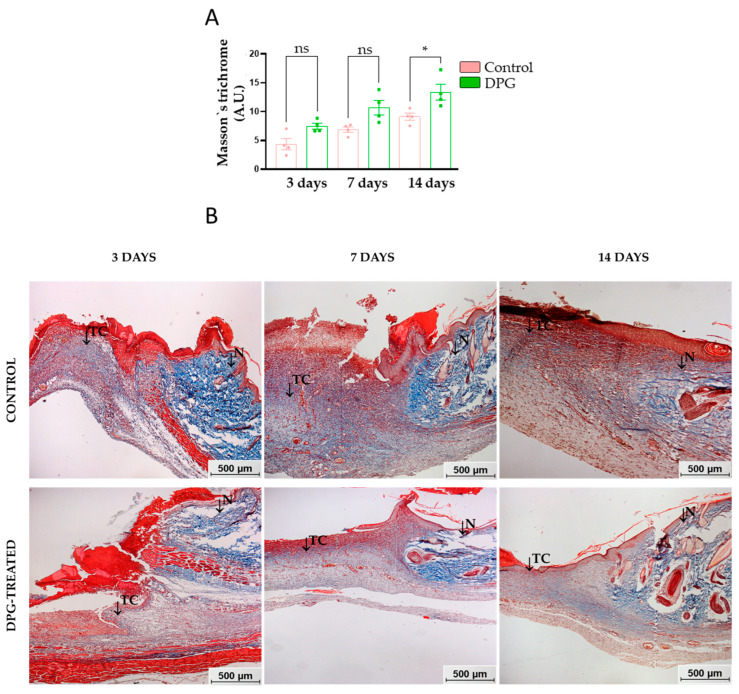
(**A**) Quantitative pixel density evaluation of total collagen content in wound scar areas over days 3, 7, and 14 of experimentation, DGP vs. Control, day 14 (* *p* < 0.05). (**B**) Representative histological images of the wound scar areas of the animals, both untreated and treated with DPG, throughout the experimentation period. Note that the presence of normal tissue (N) and the presence of total collagen (TC) is highlighted. TM staining: 50× magnification (bar = 500 μm).

**Figure 4 ijms-24-03839-f004:**
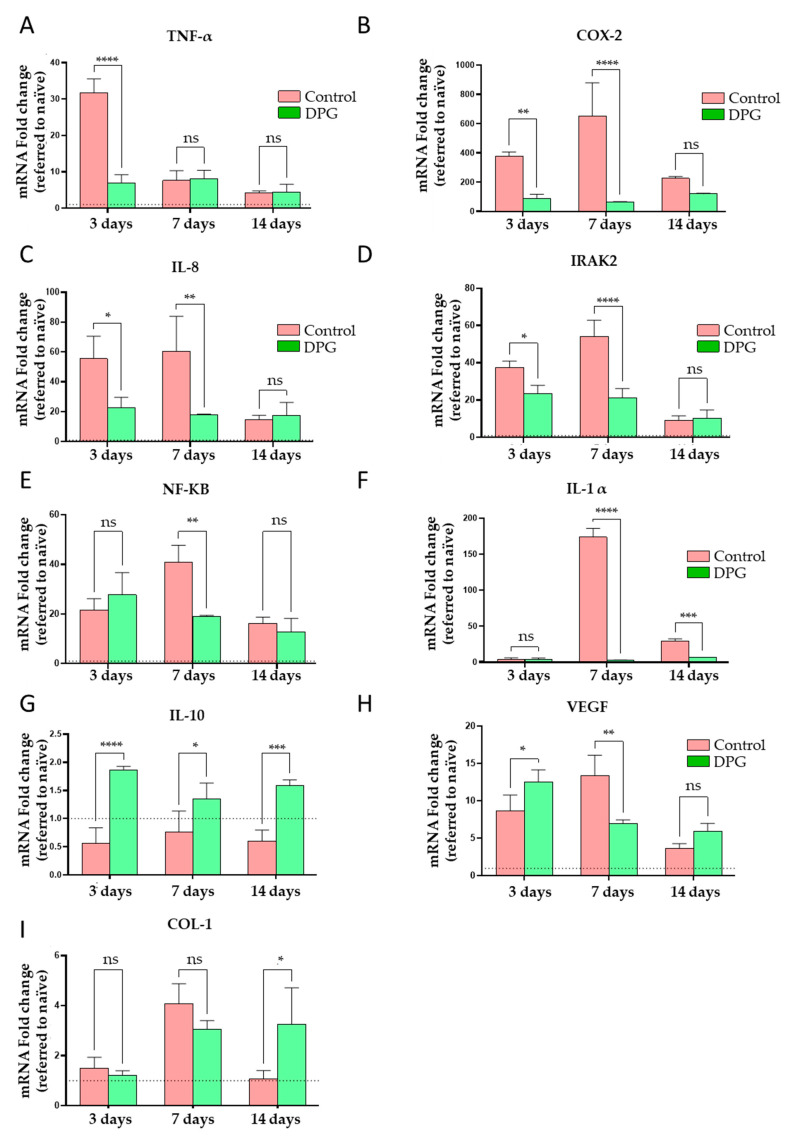
Quantitative analysis of the expression of mRNAs in the wound healing samples throughout the days of experimentation in the Control and DPG groups. (**A**) *Tnf-α* (**** *p* < 0.0001); (**B**) *Cox-2* (** *p* < 0.01, **** *p* < 0.0001)*;* (**C**) *Il-8* (* *p* < 0.05, ** *p* < 0.01); (**D**) *Irak-2* (* *p* < 0.05, **** *p* < 0.0001); (**E**) *Nf-kb* (** *p* < 0.01)*;* (**F**) *Il-1α* (**** *p* < 0.0001, *** *p* < 0.001)*;* (**G**) *Il-10* (**** *p* < 0.0001, *** *p* < 0.001); (**H**) *Vegf* (* *p* < 0.05, ** *p* < 0.01); and (**I**) *Col-1* (* *p* < 0.05). Ns—not significant, * *p* < 0.05; ** *p* < 0.01; *** *p* < 0.001; **** *p* < 0.0001 when compared to the control group.

**Figure 5 ijms-24-03839-f005:**
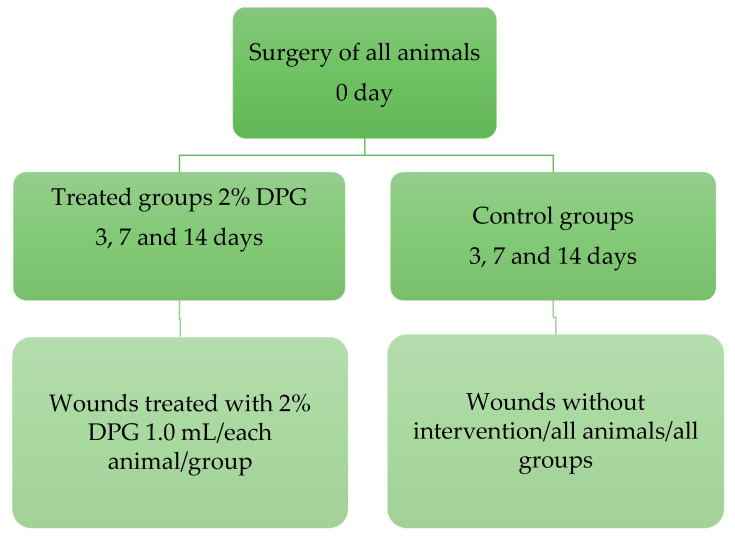
Schematic diagram of the experimental design.

**Table 1 ijms-24-03839-t001:** Composition of the dipotassium glycyrrhizinate (DPG) formulation—2% gel cream.

Ingredient	%	Function	INCI Name	CAS Number
Aqua	93.600	Solvent	Aqua	7732-18-5
Sepigel 305	4.000	Viscosity controlling	Polyacrylamide, C13-14 Isoparaffin (and) Laureth 7	9003-05-8/246538-79-4/68439-50-9/9002-92-0/7732-18-5
Dipotassium Glycyrrhizinate	2.000	Active	Dipotassium Glycyrrhizinate	68797-35-3
Euxyl PE 9010	0.400	Preservative	Phenoxyethanol (and) Ethylhexylglycerin	122-99-6/70445-33-9
Total	100.000			

**Table 2 ijms-24-03839-t002:** Inflammatory score graduation scale [134].

Degree	Score	Inflammatory Exudate	Active Hyperemia
Absent	0	No leukocyte tissue infiltration (0%)	No tissue hyperemia (0%)
Light	1	Leukocyte infiltration (<50%)	Presence of hyperemia (<50%)
Moderate	2	Leukocyte infiltration (≥51%)	Presence of hyperemia (≥51%)
Intense	3	Abundant leukocyte infiltration (≥75%)	Abundant hyperemia (≥75%)

**Table 3 ijms-24-03839-t003:** Grading scale of proliferative parameters [134].

Degree	Score	Granulation Tissue	Re-Epithelialization
Absent	0	Neoformed tissue absent (0%)	Re-epithelialization tissue absent (0%)
Light	1	Presence of neoformed tissue (<50%)	Presence of re-epithelialization tissue (<50%)
Moderate	2	Presence of neoformed tissue (≥51%)	Presence of re-epithelialization tissue (≥51%)
Intense	3	Abundant neoformed tissue (≥75%)	Presence of re-epithelialization tissue (≥75%)

**Table 4 ijms-24-03839-t004:** Sequence of primers used in qPCR.

Gene	Primers Sequence 5′-3′
*Cox-2*	FW: AACAACATTCCCTTCCTTCG RV: AAGTTGGTGGGCTGTCAATC
*Tnf-* *α*	FW: GGGCTCCCTCTCATCAGTT RV: TTGCTACGACGTGGGCTAC
*Nf-kb*	FW: CAGCTCTTCTCAAAGCAGCA RV: AGCCTTCTCCCAAGAGTCG
*Il-1* *α*	FW: GGCCATAGCCCATGATTTAG RV: TGATGAACTCCTGCTTGACG
*Irak-2*	FW: TCAAGAGGCTCAGGGAGGT RV: CCCAGCAGAGGTAGGATGTT
*Il-8rb*	FW: ATCTTTGCTGTGGTCCTCGT RV: GGTCTCCTTGATCAGCTTGG
*Il-10*	FW: AGCCTTGCAGAAAACAGAGC RV: GCCTTTGCTGGTCTTCACTC
*Vegf*	FW: CGGAGAGCAACGTCACTATG RV: GCTGCAGGAAGCTCATCTCT
*Col-1*	FW: GGAATGAAGGGACACAGAGG RV: AGGCTCTCCCTTAGGACCAG
*18s*	FW: CGCGGTTCTATTTTGTTGGT RV: CGGTCCAAGAATTTCACCTC

FW (forward), RV (reverse).

## Data Availability

Not applicable.

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
