# Peer review of "Dipotassium Glycyrrhizininate Improves Skin Wound Healing by Modulating Inflammatory Process"

_ijms, 2023, doi:10.3390/ijms24043839_

Round 1

Reviewer 1 Report

1. The language should be double checked by the authors. For example, in line 40, "through the modulation of the expression of pro-and anti-inflammatory cytokine expression;", an "expression" should be removed.

2. The quality of histological images in Figure 2 should be improved, since the current images are not clear enough for readers to check the skin repair.

3. The increase of collagen expression is related to tissue fibrosis. So, is DPG a positive drug to promote skin repair or it will induce fibrosis?

4. More markers for angiogenesis and re-epithelialization should be stain to clearly show these changes in the treated group.

Author Response

  1. The language should be double checked by the authors. For example, in line 40, "through the modulation of the expression of pro-and anti-inflammatory cytokine expression;", an "expression" should be removed.

Answer - We thank the Reviewer for the valuable comment. In the current version, the English has been corrected and edited by a native English speaker.

  1. The quality of histological images in Figure 2 should be improved, since the current images are not clear enough for readers to check the skin repair.

Answer - We thank the Reviewer for the valuable comment. All figures have had the quality increased.

  1. The increase of collagen expression is related to tissue fibrosis. So, is DPG a positive drug to promote skin repair or it will induce fibrosis?

Answer - We thank the Reviewer for the valuable comment. This issue has been addressed and inserted in the current version.

“Additionally, although the literature suggests a link between elevated collagen synthesis and the development of tissue fibrosis, this fact is really more closely linked to an escalation of the inflammatory process [129,130] and collagen production, particularly type III collagen [131,132]. We observed rapid inflammatory process resolution, improved levels of total collagen, and significant expression of type I collagen during wound remodeling, indicating maturity and resilience of the new tissue”.

  1. More markers for angiogenesis and re-epithelialization should be stain to clearly show these changes in the treated group.

Answer - We thank the Reviewer for the valuable comment. Although this is an extremely interesting question, the effects of DPG on re-epithelialization have been described previously by our group (doi: 10.1590/ACB360801). Since the scope of this manuscript is mainly about the anti-inflammatory activity of DPG, we respectfully cannot fulfill the reviewer's request at this time.

Reviewer 2 Report

The authors aimed to evaluate the effects of dipotassium glycyrrhizinate (DPG) on wound healing in rats after 3, 7 and 14 days of treatment, addressing the involvement of pro- and anti-inflammatory factors, considering the lack of reports on the effect of DPG on cutaneous  inflammatory processes during wound healing.

 They conclude that DPG attenuates the inflammatory process by promoting skin wound healing through the modulation of distinct mechanisms and signaling pathways, including anti-inflammatory, through the modulation of the expression of pro-and anti-inflammatory cytokine expression, promotion of new granulation tissue, angiogenesis and tissue re-epithelialization contributing to tissue remodeling.

The manuscript is well structured and the methods and data are quite clearly presented, but it requires some improvements:

Perhaps the authors should introduce a phrase about licorice (scientific name, occurrence), even if the plant has been intensively studied, as well as some data about the percentage of Glycyrrhizin (G) in the plant.

The authors should specify more clearly the scientific contribution of the work, not only ‘considering the lack of reports’, considering that they have two recent articles on the same topic.

In all figures, the diagrams are unclear, even at considerable magnification.

A review of English (spelling and grammar) is necessary.

Author Response

The authors aimed to evaluate the effects of dipotassium glycyrrhizinate (DPG) on wound healing in rats after 3, 7 and 14 days of treatment, addressing the involvement of pro- and anti-inflammatory factors, considering the lack of reports on the effect of DPG on cutaneous inflammatory processes during wound healing.

They conclude that DPG attenuates the inflammatory process by promoting skin wound healing through the modulation of distinct mechanisms and signaling pathways, including anti-inflammatory, through the modulation of the expression of pro-and anti-inflammatory cytokine expression, promotion of new granulation tissue, angiogenesis and tissue re-epithelialization contributing to tissue remodeling.

  1. The manuscript is well structured and the methods and data are quite clearly presented, but it requires some improvements: Perhaps the authors should introduce a phrase about licorice (scientific name, occurrence), even if the plant has been intensively studied, as well as some data about the percentage of Glycyrrhizin (G) in the plant.

Answer - We thank the Reviewer for the valuable comment. This information was added in the current version of the manuscript.

  1. The authors should specify more clearly the scientific contribution of the work, not only‘considering the lack of reports’, considering that they have two recent articles on the same topic.

Answer - We thank the Reviewer for the valuable comment. The text was restructured so that the relevance of the article was highlighted. “Although a recent study found that rats treated with DPG experienced effective epidermal proliferation and re-epithelialization and an increase in type I collagen levels [31], the mechanism of action of this chemical on other crucial stages of wound healing has hardly been investigated. Thus, this work aimed to evaluate the effects of DPG on wound healing in rats after 3, 7 and 14 days of treatment, addressing the involvement of pro- and anti-inflammatory factors”

3. In all figures, the diagrams are unclear, even at considerable magnification.

Answer - We thank the Reviewer for the valuable comment. All figures have had the quality increased.

  1. A review of English (spelling and grammar) is necessary.

Answer - We thank the Reviewer for the valuable comment. In the current version, the English has been corrected and edited by a native English speaker.

Reviewer 3 Report

The research paper entitled “Glycyrrhizininate improves skin wound healing by modulating inflammatory process (Manuscript ID: ijms-2141379)” was reviewed. After reading the manuscript, I suggest to author to revise wisely for publication in International Journal of Molecular Science. Please do the revision for this manuscript based on comments below (major revision). This manuscript is not acceptable in this format.

1. The abstract should be re-written to summarize the work; the abstract should state briefly the purpose of the research, the PRINCIPLE results and MAJOR conclusions. An abstract is often presented separately from the article, so it must be able to stand alone

2. The novelty of this research article compared to other studies is very low. There are many researches published in recent years about magnetic materials and using them for catalytic application.

3. Authors must compare their results (in a results and discussion section) with others reported in the literature.

4. Redesign the methods chapter the way so anybody can repeat your procedures, like a recipe.

5. The quality and explain of Fig.1 is very low.

6. The mechanism of wound treatment is not clear. Please more explain and explain schematically.

7. The authors must revise the manuscript carefully to eliminate grammatical errors and typo-errors.

Author Response

The research paper entitled “Glycyrrhizininate improves skin wound healing by modulating inflammatory process (Manuscript ID: ijms-2141379)” was reviewed. After reading the manuscript, I suggest to author to revise wisely for publication in International Journal of Molecular Science. Please do the revision for this manuscript based on comments below (major revision). This manuscript is not acceptable in this format.

  1. The abstract should be re-written to summarize the work; the abstract should state briefly the purpose of the research, the PRINCIPLE results and MAJOR conclusions. An abstract is often presented separately from the article, so it must be able to stand alone.

Answer - We thank the Reviewer for the valuable comment. The abstract has been restructured properly..

  1. The novelty of this research article compared to other studies is very low. There are many researches published in recent years about magnetic materials and using them for catalytic application

Answer - We thank the Reviewer for the valuable comment. We understand the reviewer's point. The novelty of the paper lies in the better understanding of the anti-inflammatory effects of DPG improving skin wound healing. We believe that in the updated version of the article, the relevance of the article has become more evident.

  1. Authors must compare their results (in a results and discussion section) with others reported in the literature.

Answer - We thank the Reviewer for the valuable comment. As required, in the current version of the manuscript, a more comprehensive comparison with literature was included.

  1. Redesign the methods chapter the way so anybody can repeat your procedures, like a recipe.

Answer - We thank the Reviewer for the valuable comment. The M&M section was redesigned.

  1. The quality and explain of Fig.1 is very low.

Answer - We thank the Reviewer for the valuable comment. All figures have had the quality increased.

  1. The mechanism of wound treatment is not clear. Please more explain and explain schematically.

Answer - We thank the Reviewer for the valuable comment. The wound treatment was explained in deeper detail.

  1. The authors must revise the manuscript carefully to eliminate grammatical errors and typo-errors.

Answer - We thank the Reviewer for the valuable comment. In the current version, the English has been corrected and edited by a native English speaker.

Reviewer 4 Report

The authors have studied the effect of Glycyrrhizininate in accelerating skin wound healing by modulating inflammatory process. The study design and methodology are sound. However, I have a major concern regarding the novelty of the research, since there is vast array of literature available on the wound healing potential of Glycyrrhizininate (Refer the following articles: doi: 10.1590/ACB360801; https://doi.org/10.1016/j.actbio.2022.02.041).

I have the following comments:

1. The authors must clarify the significance of the central claims of the present study in the context of the existing literature, what the present study adds to what was already done. And what gaps are the authors trying to fill-in?  For example, in the introduction the authors must put stress on the molecular mechanisms of wound healing which have not been reported earlier for glycyrrhizinate.

2. The discussion can be improved by citing similar previous studies and comparing your results with the earlier studies and focusing on the antiinflammatory mechanisms of glycyrrhizinate which is evident from the microscopic observations and reverse transcription quantitative PCR (qPCR) data from this study.

3. In methodology: Section 3.4. Topical Treatment: The Wounds in the treated groups (DPG) were treated once a day for 3, 7 and 14 days, at the same time and by the same researcher, with topical application of 2% DPG cream (0.1 mL/animal).

The authors have not mentioned how the cream/ointment were prepared?

The methodology for the formulation of cream must be clearly mentioned with references. Please refer: https://doi.org/10.1155/2022/6449550

4. The font in the figures is very small. The figure clarity must be improved and font size increased.

5. Please revise the manuscript for English and grammar.   

Author Response

  1. The authors have studied the effect of Glycyrrhizininate in accelerating skin wound healing by modulating inflammatory process. The study design and methodology are sound. However, I have a major concern regarding the novelty of the research, since there is vast array of literature available on the wound healing potential of Glycyrrhizininate (Refer the following articles: doi: https://doi.org/10.1590/ACB360801; https://doi.org/10.1016/j.actbio.2022.02.041).

I have the following comments:

  1. The authors must clarify the significance of the central claims of the present study in the context of the existing literature, what the present study adds to what was already done. And what gaps are the authors trying to fill-in? For example, in the introduction the authors must put stress on the molecular mechanisms of wound healing which have not been reported earlier for glycyrrhizinate.

Answer - We thank the Reviewer for the valuable comment. In the current version of the manuscript, the work's relevance has been highlighted.

  1. The discussion can be improved by citing similar previous studies and comparing your results with the earlier studies and focusing on the antiinflammatory mechanisms of glycyrrhizinate which is evident from the microscopic observations and reverse transcription quantitative PCR (qPCR) data from this study.

Answer - We thank the Reviewer for the valuable comment. The discussion has been improved by comparing it more with the literature. In this way, we believe that the article becomes more appealing.

  1. In methodology: Section 3.4. Topical Treatment: The Wounds in the treated groups (DPG) were treated once a day for 3, 7 and 14 days, at the same time and by the same researcher, with topical application of 2% DPG cream (0.1 mL/animal).

Answer - We thank the Reviewer for the valuable comment. Yes, all experimental data (animal data) were conducted by the same investigator.  

The authors have not mentioned how the cream/ointment were prepared? The methodology for the formulation of cream must be clearly mentioned with references. Please refer: https://doi.org/10.1155/2022/6449550.

Answer - We thank the Reviewer for the valuable comment. This information was added in the current version of the manuscript .

Round 2

Reviewer 3 Report

Accept

Reviewer 4 Report

The authors have done a reasonable effort to revise the manuscript. 

The manuscript appears to be more appealing to the readers after the revisions.